# The Dynamic Change in Fatty Acids during the Postharvest Process of Oolong Tea Production

**DOI:** 10.3390/molecules27134298

**Published:** 2022-07-04

**Authors:** Zi-Wei Zhou, Qing-Yang Wu, Yun Yang, Qing-Cai Hu, Zong-Jie Wu, Hui-Qing Huang, Hong-Zheng Lin, Zhong-Xiong Lai, Yun Sun

**Affiliations:** 1College of Life Science, Ningde Normal University, Ningde 352000, China; zwchow92@126.com; 2Key Laboratory of Tea Science in Fujian Province, College of Horticulture, Fujian Agriculture and Forestry University, Fuzhou 350002, China; doris1831036881@126.com (Q.-Y.W.); fafuyy@163.com (Y.Y.); hqcxuexi@163.com (Q.-C.H.); fafuwzj@163.com (Z.-J.W.); hhq9959@163.com (H.-Q.H.); linhongzheng2010@126.com (H.-Z.L.); 3Institute of Horticultural Biotechnology, Fujian Agriculture and Forestry University, Fuzhou 350002, China; laizx01@163.com

**Keywords:** oolong tea, manufacturing process, fatty acids, LOX

## Abstract

As important factors to oolong tea quality, the accumulation and dynamic change in aroma substances attracts great attention. The volatile composition of oolong tea is closely related to the precursor contents. Fatty acids (FAs) and their derivatives are basic components of oolong tea fragrance during the postharvest process. However, information about the precursors of FAs during the postharvest process of oolong tea production is rare. To investigate the transformation of fatty acids during the process of oolong tea production, gas chromatograph–flame ionization detection (GC-FID) was conducted to analyze the composition of FAs. The results show that the content of total polyunsaturated FAs initially increased and then decreased. Specifically, the contents of α-linolenic acid, linoleic acid and other representative substances decreased after the turn-over process of oolong tea production. The results of partial least squares discrimination analysis (PLS-DA) showed that five types of FAs were obviously impacted by the processing methods of oolong tea (VIP > 1.0). LOX (Lipoxygenase, EC 1.13.11.12) is considered one of the key rate-limiting enzymes of long-chain unsaturated FAs in the LOX-HPL (hydroperoxide lyase) pathway, and the mechanical wounding occurring during the postharvest process of oolong tea production greatly elevated the activity of LOX.

## 1. Introduction

Fatty acids (FAs) are key constituents of all plant cells, providing structural integrity and energy for various metabolic processes and functioning as signal transduction mediators [1]. In fresh tea leaves, FAs are degraded into volatile components after polyinsaturation or ployunsaturation [2,3]. Based on the metabolic differences of hydroperoxide (9/13-HPOT), the oxidation product of the key rate-limiting enzyme LOX, the LOX pathway could be divided into the LOX-AOS branch and LOX-HPL branch [4,5]. In the former branch, 13S-HPOT could respond to abiotic stress as well as be degraded to 12-oxophytodienoic acid with the help of allene oxide synthase (AOS) and allene oxide cyclase (AOC). Besides, 12-oxophytodienoic acid undergoes one time of reduction and three times of β-oxidation reactions to form jasmonic acid (JA), which is one of the fastest signal substances in the adversity during the process of plant growth and development [6]. As for the latter branch, it mainly responds to exogenous biotic and abiotic stresses. Therefore, green leaf volatiles (GLVs) and fruit esters with diffuse aroma such as trans-2-hexenal, *cis*-3-hexenol, *cis*-3-hexenyl acetate and hexyl hexanoate are formed progressively under the oxidation, reduction and esterification of HPL, ADH as well as AAT [2,7] (Figure 1).

Previous research mainly focused on the composition and content of FAs of tea plants, including fresh tea leaves in different seasons [8], green tea [9,10], black tea [11], oolong tea [12], dark tea [13] and store processes of made tea [14]. To sum up, it is considered that the long-chain unsaturated fatty acids such as α-linolenic, linoleic, palmitic and oleic acid are the main origin of fatty fragrance in processed tea [15].

As one of the six traditional Chinese tea types, oolong tea is famous for its natural fragrance of flowers and fruit. The aroma of oolong tea is a comprehensive manifestation of tea varieties [16], geographical environment [17] and manufacturing processes [18]. In detail, tea varieties lay the groundwork for the aroma of oolong tea. The regional environment highlights the aroma and quality of oolong tea. The manufacturing process is the basic guarantee for the aroma formation of oolong tea. According to the metabolic origin, tea volatile compounds can be divided into four major classes: fatty acid derivatives, phenylpropanoids/benzenoids, terpenoids and norisoprenoids [19,20]. Among these classes, volatile compounds derived from fatty acids have undergone a series of profound changes during the oolong tea manufacturing process. However, there have been no reports on the diversification and dynamics of FAs during the postharvest process of oolong tea production so far. Therefore, with a strong purpose to know about the fatty acid changes and their response to mechanical force during the postharvest process of oolong tea production, we used GC-FID with Supelco^®^ 37 Component Fatty acid methyl esters (FAME) Mix to analyze the component and the content of fatty acids in processing oolong tea samples. Additionally, UPLC-MS/MS was also applied in detecting the precursor substance of fatty acids, which includes linolenic, linoleic, palmitic and trans-9-oleic acid. This article contributes to both the oolong tea processing technique and quality improvement.

## 2. Results

### 2.1. Qualitative and Quantitative Analysis of FAs by GC-FID

The GC-FID was applied to determine Supelco 37 Component FAME Mix as shown in Appendix A, and 21 FAMEs were distinguished based on retention time in Table 1.

As a result, the saturated fatty acids (SFAs), monounsaturated fatty acids (MUFAs) and polyunsaturated fatty acids (PUFAs) were quantitative in the fresh tea leaves (Figure 2). The contents of PUFAs were the most abundant in tea leaves (1878.54 μg·g^−1^, three compounds), followed by SFAs (830.82 μg·g^−1^, nine compounds) and MUFAs (356.84 μg·g^−1^, three compounds).

### 2.2. Dynamic Change in FAs Component during the Postharvest Process of Oolong Tea Production

The tea leaves in different processing steps were collected to monitor the dynamic changes of FAs over the oolong tea production. As shown in Figure 3, various alteration trends of FAs were observed during the oolong tea production process. The contents of SFAs decreased in withered leaves (W), then sharply increased in the leaves after turn-over (T3) compared with those in fresh leaves (F). Five individual SFAs, including capric acid (C10:0), undecanoic acid (C11:0), lauric acid (C12:0), myristic acid (C14:0), palmitic acid (C16:0) and methyl stearate (C18:0) showed an overall rising trend over the process of oolong tea production, of which the undecanoic acid was only detected in turn-over tea leaves (0.06%). Notably, palmitic acid was the most abundant SFA in tea leaves over oolong tea production, whose overall change trend could reflect the total content of SFAs, especially during the turn-over stage (Figure 3G).

As presented in Figure 4, the total content of MUFAs increased 0.27% after withering and subsequently decreased 1.32% after turn-over.

In detail, the contents of etradecaenoic acid (C14:1) and *cis*-10-pentaenoic acid (C15:1) demonstrated overall rising trends, while the other MUFAs showed opposite trends along with oolong tea processing (Figure 4). The most abundant compounds in the processing tea leaves were trans-9-oleic acid (C18:1t), whose contents were barely changed in the withering step (10.26%), then rapidly declined to 9.03% (Figure 4e), suggesting a significant conversion over the turn-over step.

The variation trend of total PUFAs content was comparable to that of MUFAs during the oolong tea production (Figure 5). As the most abundant compound in PUFAs, α-linolenic acid (C18:3(n-3)) rose slightly to 41.12% after withering, then decreased significantly after turn over (36.30%). As for other MUFAs, γ-linolenic acid (C18:3(n-6)), *cis*-5,8,11,14,17-eicosapentaenoic acid (C20:5(n-3)) and *cis*-13,16-docosadienoic acid(C22:2) were increased over the process of oolong tea production (Figure 5).

### 2.3. Influence of Mechanical Wounding on FAs Component

To investigate the fatty acid changes in the tea leaves subjected to mechanical wounding occurring in the turn-over process, the oolong tea leaves in the T3 and CK groups were collected and applied to the determination, respectively.

The results are shown in Figure 6. The total content of SFAs in turn-over tea leaves (30.09%) was higher than the same time spreading tea leaves. In detail, capric acid (C10:0), undecanoic acid (C11:0), lauric acid (C12:0), myristic acid (C14:0), palmitic acid (C16:0) and stearic acid (C18:0), which all belong to SFAs, shared the same pattern and, notably, the content of palmitic acid in turn-over tea leaves (23.67%) was significantly higher than spreading tea leaves (22.36%). On the contrary, the content of ultra-long chain SFAs such as arachidic acid (C20:0), behenic acid (C22:0), xylic acid (C24:0) and pentadecanoic acid (C15:0) were opposite, among which the content of xylic acid in turn-over tea leaves (0.49%) was significantly lower than spreading tea leaves (0.63%).

As shown in Figure 7, compared with spreading tea leaves, the total content of MUFAs in turn-over tea leaves decreased 0.25%, and four kinds of long chain FAs such as palmitoleic acid (C16:1), trans-9-oleic acid (C18:1t), *cis*-eicosenoic acid (C20:1) and erucic acid (C22:1) should be mentioned as they also decreased; the palmitoleic acid content in turn-over tea leaves (0.29%) especially was significantly lower than spreading tea leaves (0.44%). However, the change in tetradecaenoic acid (C14:1) and *cis*-10-pentaenoic acid (C15:1) content was in an opposing direction.

Surprisingly, the only extremely significant difference of FA content between turn-over and spreading tea leaves belongs to the total content of PUFAs, which is the principle kind of FA in fresh tea leaves. In detail, the total content of PUFAs in turn-over tea leaves (57.42%) was lower than spreading tea leaves (57.42 + 4.20 = 61.62%), and this pattern also could be found in linoleic acid (C18:2(n-6)), α-linolenic acid (C18:3(n-3)) and *cis*-13,16-docosadienoic acid (c22:2). Acting as representative substance in PUFAs, the content of α-linolenic acid in turn-over tea leaves (36.30%) was extremely significantly lower than the spreading tea leaves (40.34%). Compared with its isomer α-linolenic acid, the proportion of γ-linolenic acid was much lower in PUFAs, and its content in turn-over and spreading tea leaves was very close (0.053% and 0.052%, respectively). Combined with the former detection results on fresh and withered tea leaves, *cis*-5,8,11,14,17-eicosapentaenoic acid (C20:5(n-3)) also cannot be found in spreading tea leaves; therefore, we thought this substance may only respond to mechanical force (Figure 8).

### 2.4. Recognition and Cluster Analysis of Significant Differential Compounds

In order to estimate the component of key FAs during the different processes of oolong tea production, the dataset obtained from GC-FID was applied to make an analysis using a partial least squares discrimination analysis (PLS-DA) model on SIMCA P 14.1. Clearly, the triplicates of each sample were gathered and discriminated from other samples (Figure 9a). Next, to refine the significantly different metabolites, the variable importance in projection (VIP) values were calculated from the PLS-DA model. As shown in Figure 9b, a total of five FAs, including behenic acid (C22:0), α-linolenic acid (C18:3(n-3)), palmitic acid (C16:0), stearic acid (C18:0) and trans-9-oleic acid (C18:1), were obtained as the significant differential compounds with the criteria of VIP value exceeding 1.0. A permutation test was performed to assess model fitting by iterating 200 times. The cross-validation results indicated no overfitting in the model (intercepts, R2 = 0.134 and Q2 = −0.36, Figure 9c).

For the purpose of presenting the component of five kinds of FAs during the postharvest process of oolong tea production, hierarchical clustering (HCL) was applied in analysis with color degree closer to orange, stressing the higher content, and when closer to the color green, the content was much lower. As the Figure 9d shows, five kinds of FAs were divided into three main clusters: the first kind related to higher proportions of α-linolenic acid and palmitic acid, which showed opposite trends during the postharvest process of oolong tea production, especially in T3; the content of palmitic acid reached maximum 36.30% and α-linolenic acid minimum 23.67%. The second cluster belongs to trans-9-oleic acid and stearic acid, both also showed opposite trends with minimum 9.03% and maximum 3.61%, respectively, in T3. The final cluster, which occupied the lowest proportion, showed decreasing trends from maximum 1.64% in fresh tea leaves to minimum 0.30% in T3.

### 2.5. Component Measurement of Main FAs during Postharvest Process of Oolong Tea by UPLC-MS/MS

As mentioned in previous results, the total contents of α-linolenic acid (C18:3(n-3)) (40.45%), palmitic acid (C16:0) (20.63%), linoleic acid (C18:2(n-6)) (20.64%) and trans-9-oleic acid (c18:1t) (10.12%) were more than 90%, and these four kinds of FAs all had significant or extremely significant differences during the postharvest process of oolong tea production. Furthermore, these four substances have proved to be key precursors of volatile fatty acid derivatives (VFADs) as well [15]. It is worth noting that the α-linolenic acid and palmitic acid both responded to the mechanical force of turn over significantly, what is more, the model of PLS-DA declared that α-linolenic acid, palmitic acid and trans-9-oleic acid all were practical in testing and distinguishing the key components of FAs in processing oolong tea. Therefore, UPLC-MS/MS was applied for qualitative and quantitative detection of these four kinds of FAs in order to explicitly find out the relative trend during the postharvest process of oolong tea production and further confirm the testing results of GC-FID. Next, electrospray ionization (ESI) under negative ion scanning combined with multiple reaction monitoring (MRM) analysis was used for precise sensitivity. There also followed optimized mass spectrum parameters such as declustering potential (DP) and collision energy (CE), which were ideal for rapid screening and determination of the target compound. The mass spectrum conditions of the four target compounds were given in Appendix A. Finally, the chromatograms and standard curves of oolong tea processing samples (F, T3, CK3) are shown in Figure 10.

Compared with the chemical reference substance (CRS), it was found that all samples obtained a distinct peak without few miscellaneous peaks, and in the concentration range of 600~20,000 ng/mL, the peak intensity of the standard related closely with the concentration. Meanwhile, the retention time standard deviation range of palmitic acid, trans-9-oleic acid, α-linolenic acid and linoleic acid remains around 0.94~1.48%, indicating that the optimized analysis method mentioned above has the characteristics of higher sensitivity and efficiency, also, less impurity interference.

UPLC-MS/MS was applied in testing the content of palmitic acid, trans-9-oleic acid, α-linolenic acid and linoleic acid in fresh tea leaves(F), turn-over tea leaves (T3) and spreading tea leaves (CK3). The results show that in turn-over tea leaves (T3), the content of palmitic acid (0.40%) is extremely higher than fresh tea leaves (0.29%) and significantly higher than spreading tea leaves (0.32%). As for α-linolenic acid, its content between all three groups had significant differences (0.30% in F, 0.28% in CK3), and the lowest occurred in T3 (0.22%) (Appendix A). Although similar to the trend of α-linolenic acid, the highest content of linoleic acid occurred in spreading tea leaves (0.19%) and without significant difference between turn-over and fresh tea leaves. However, there were no significant differences between all treatments in trans-9-oleic acid. Comparing both testing results of GC-FID and UPLC-MS/MS, we found that the content of palmitic acid (R = 0.863, *p* < 0.01) and α-linolenic acid (R = 0.800, *p* < 0.01) has an extremely significant relationship, and linoleic acid has a significant relationship (R = 0.699, *p* < 0.05), while trans-9-oleic acid has no significant relationship (*p* < 0.05) (Table 2).

### 2.6. Change in LOX Activity during the Postharvest Process of Oolong Tea Production

LOX is a key rate-limiting enzyme that exclusively catalyzes long-chain FAs in the LOX-HPL metabolic pathway. As shown in Figure 11, the activity of LOX increased along with the process of oolong tea production, ranging from 4080 U/g in fresh leaves to 10,080 U/g. Moreover, the activity of LOX in CK was only 6400 U/g, showing a negligible change compared with that in fresh leaves. This observation suggested that the mechanical wounding occurring in the turn-over step enhanced the activity of LOX in response to the stress, in contrast to the unchanged LOX activity in CK.

## 3. Discussion

### 3.1. Content of FAs in Fresh Oolong Tea Leaves

In this study, a total of 21 FAs were identified and quantified in oolong tea leaves, most of which were unsaturated FAs. This observation is in line with the previous study that the fresh leaves of the tea plant were mainly composed of unsaturated FAs [21], and the main FA components in the fresh leaves of *C. sinensis* cv. Longjingchangye were α-linolenic acid (46.47%) and linoleic acid (16.63%) [22]. The proportion of α-linolenic acid and linoleic acid accounted for 61.07%, while trans-9-oleic acid was only 10.12% and the remainder just less than 1%. Therefore, we could conclude that the type of FAs in fresh oolong tea leaves were long-chain FAs with 18 carbon and 16 carbon, and there were sure differences between the content of varied FAs. With the increasing order of tea leaf position, the content of FAs increased [8]. To some extent, this phenomenon may explain why fresh, mature oolong tea is rich in FAs. As for the relationship between maturity degree and FA content of tea leaves, based on the previous studies, a higher degree contributes to thicker waxy layers, which are made of very long-chain FAs (VLCFAs) and their relative derivatives [23,24]. However, in our study, in fresh oolong tea leaves, six kinds of VLCFAs content, including arachidonic acid (0.44%), xylic acid (0.56%) and behenic acid (1.64%) in SFAs, *cis*- eicosenoic acid (0.34%) and erucic acid (0.28%) in MUFAs and *cis*-13,16-docosadienoic acid (0.29%), were all lower. Previous studies have proved that the leaf epidermis composed of cuticle and wax layer was located outside the plant epidermal cells [25], and in Southern Fujian’s tea varieties, the cuticle mainly appeared in the third and fourth leaves under microstructure comparison [26]. Therefore, when concerning the reasons mentioned for surprising experimental results, speculations were conveyed that the thickness and content of waxiness in one bud and three leaves were both less than matured tea leaves as well as the thin leaf materials collected from C. *sinensis* cv. Huangdan and may be attributed to a lower content of VLCFAs.

### 3.2. Change in FAs during the Postharvest Process of Oolong Tea Production

FAs are an important precursor substance of tea fragrance, accounting for nearly 1% of the dry weight of fresh tea leaves [27,28]. The content of PUFAs showed a decreasing trend along with the process of oolong tea production, which was similar to the alteration of MUFAs. By contrast, the content of SFAs increased along with the process of oolong tea production. Therefore, the FAs may divide into short chain from long chain with gradual saturation during the postharvest process of oolong tea production. Based on the differences in the content of FAs components in the samples, we established a PLS-DA model to preliminarily distinguish samples during the postharvest processing of oolong tea production [29]. According to the discrete degree of F_v_T3 and F_v_CK3 in the load diagram, we could infer that the participation of the mechanical force is an important factor in promoting the change in the principal components of FAs [30]. The FAs screened under the condition that vip >1.0 reflected that the change in fatty acid carbon chain is more obvious than the change in saturated bond [31,32]. The activity of LOX in turn-over tea leaves was significantly higher than that in spreading leaves, suggesting that the LOX were significantly enhanced by the mechanical wounding. This result is consistent with the dynamic trend of LOX activity during the manufacturing process of Guangdong oolong tea [33] and the processing technology affecting FAs composition of oolong tea production [34]. Furthermore, it was found that the plant cell membrane, composed of lipids, plays an important role in response to drought and mechanical injury [35]. However, a similar response was the stimulating of LOX firstly, then the stimulated LOX oxidized and split the unsaturated FAs during the postharvest process of oolong tea production. Afterwards, with the help of HPL from the downstream, the octadecane unsaturated fatty acid, which is represented by α-linolenic acid, split at the sites of the 9th and 13th double bonds, respectively After the oxidation effect happened at the sites of the 13th, the eighteen carbon unsaturated FAs broke down into six carbon volatile substances and twelve carbon saturated FAs; among these two, the former was the key origin of delicate fragrance in tea, and the latter was related to the accumulation of total SFAs during the postharvest process of oolong tea production. Last but not least, based on the testing of GC-FID and UPLC-MS/MS, we could confirm that along with the oolong tea process, the activity of LOX grew, the polyunsaturated FAs represented by α-linolenic acid and linoleic acid decreased, while the saturated FAs represented by palmitic acid increased; both results are the same as Lin YF’s experiment conveyed in the postharvest of Longan [36]. Therefore, derivatives of α-linolenic acid and linoleic acid through the LOX-HPL pathway lay the foundation for the formation of fragrance of oolong tea, directly or indirectly.

## 4. Materials and Methods

### 4.1. Materials

Tea (*Camellia sinensis* cv. Huangdan) shoots were acquired in October 2018 from the tea plantation (26°04′ N, 119°14′ E) of Fujian Agriculture and Forestry University (Fuzhou, China). Tea shoots composed of three leaves and a bud were applied to produce oolong tea in this study.

### 4.2. Methods

#### 4.2.1. Reagents

Isopropanol, chloroform, methanol, potassium chloride, sulfuric acid and toluene all were analytically pure (Sinopharm Chemical Reagent Co., Ltd., Shanghai, China). N-hexane, Supelco 37 Component FAME Mix (Appendix A), α-linolenic, linoleic, palmitic, oleic acid and other referencing standards were chromatographically pure (Sigma, Saint Louis, MO, USA). LOX activity detection kit (Beijing Solarbio Science & Technology Co., Ltd., Beijing, China) was used for the measurement of enzyme activity in oolong tea leaves.

#### 4.2.2. Postharvest Treatments and Samples Preparation

As shown in Figure 12, fresh tea leaves (F) were subjected to solar withering under sunlight (120,000 Lux, 25 °C) for 30 min, then the withered leaves were separated as a turn-over group (T3), collected after a three-phase turn over, and the other control group was standing still (CK3). Both of these two treatments were maintained in the same workshop under the same conditions of ambient temperature (24 °C) and moisture (45%). All sampling for each treatment above was repeated at least three times and wrapped in tin foil, then immediately immersed in liquid nitrogen for 3 min, then placed in dry ice for transportation prior to storing at −70 °C.

#### 4.2.3. Fatty Acid Content Determination via Direct Methylation

Dried tea leaves (400 mg) were weighed on an analytical balance. A glass tube (1 cm × 10 cm) with Teflon-lined screw cap was prerinsed thoroughly with chloroform and dried to remove any contaminating lipid residues. To this tube was added 1 mL of 5% (*v*/*v*) conc. sulfuric acid in MeOH, 25 μL of BHT solution (0.2% butylated hydroxy toluene in MeOH), 10–100 μL of triheptadecanoin (as a triacylglycerol internal standard to generate methyl heptadecanoate) and 300 μL of toluene as cosolvent. The mixture was vortexed for 30 s then heated at 95 °C for 1 h. After cooling to room temperature, 2 mL of 0.9% NaCl (*w*/*v*) was added, and FAMEs were extracted with 1 mL hexane. Pooled extracts were evaporated under nitrogen and then dissolved in 400 μL of hexane. The FAME extracts were analyzed by GC with a flame ionization detector (FID) on a DB23 column (30 m by 0.25 mm i.d., 0.25 μm film; J&W Scientific, Folsom, CA, USA). The GC conditions were: split mode injection (1:20), injector and flame ionization detector temperature 280 °C; oven temperature program 150 °C for 3 min, then increasing at 10 °C/min to 230 °C and holding this temperature for 8 min. In preliminary experiments, a time course of the transmethylation reaction was generated using squalane and triheptadecanoin as internal standards.

#### 4.2.4. Calculation of Fatty Acid Content from GC Analysis

The area of each peak in the GC chromatogram was first corrected for the theoretical response factor of the FID in which peak area (pA.sec) is a function of mass of C atoms with at least one bound H atom, and which therefore differs for each fatty acid methyl ester [37]. The corrected areas were used to calculate the mass of each FAME in the sample by comparison with the internal standard mass. Because fatty acid stored in tea leaves is primarily in the form of triacylglycerol, an additional correction to convert FAME weight to TAG weight is needed. This correction depends on the fatty acid composition of the oil. To determine the moles of TAG, the moles of each FAME were calculated from the weight and the molecular weight. Calculation of sample concentration: substitute the mass spectrum peak area of the sample analysis into the linear equation to calculate the concentration. Percent of fatty acid by weight = (C * V1)/M × 10 − 6. Among them, C stands for the concentration measured by LC-MS; V1 stands for the volume of 1 mL before loading and M stands for the weight of the sample during extraction.

#### 4.2.5. LOX Activity Analysis

The determination of LOX activity was performed by using a LOX activity detection kit following the instructions of the manufacturer (Beijing Solarbio Science & Technology Co., Ltd., Beijing, China). Briefly, frozen samples (0.1 g) were milled into powder in liquid nitrogen, then suspended in 1 mL of extraction solution at 4 °C. Then, the solution was centrifuged at 16,000× *g* at 4 °C for 20 min; the supernatant was subjected to spectrophotometer detection.

#### 4.2.6. Statistical Analysis

All experimental results are provided as mean ± standard error of the mean. Statistical analysis was conducted using SPSS (PASW Statistics Base 18, IBM, Chicago, IL, USA) to determine significance. Statistical differences between measurements were assessed using Tukey’s test (* *p* < 0.05 and ** *p* < 0.01) after an analysis of variance. All figures were produced using Prism (GraphPad, Version 6.01, GraphPad Software Inc., San Diego, CA, USA).

## 5. Conclusions

This study analyzed the dynamic changes of FAs during the process of oolong tea production. The difference in postharvest treatment of oolong tea leaves might affect the metabolic process over the LOX pathway, potentially contributing to the accumulation of several biomarkers such as behenic acid, α-linolenic acid, palmitic acid, stearic acid and trans-9-oleic acid. The results consolidate that the mechanical wounding is the crucial factor that induces the characteristic compounds during the production of oolong tea. Turn over is an effective means to improve the aroma of tea leaves. Appropriate turn-over treatment could promote the degree of fatty acid oxidation, enhancing the aroma of summer tea, which improves the economy of tea plucked in the summer season.

## Figures and Tables

**Figure 1 molecules-27-04298-f001:**
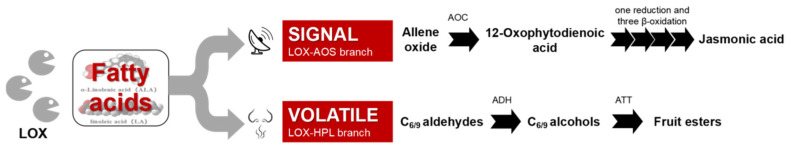
Two branches of fatty acid metabolic pathways in plant. LOX, Lipoxygenase (EC 1.13.11.12); AOS, allene oxide synthase (EC 4.2.1.92); AOC, allene oxide cyclase (EC 5.3.99.6); HPL, hydroperoxide lyase (EC 4.1.2.−); ADH, alcohol dehydrogenase (EC 1.1.1.1); AAT, alcohol acyltransferase (EC 2.3.1.−).

**Figure 2 molecules-27-04298-f002:**
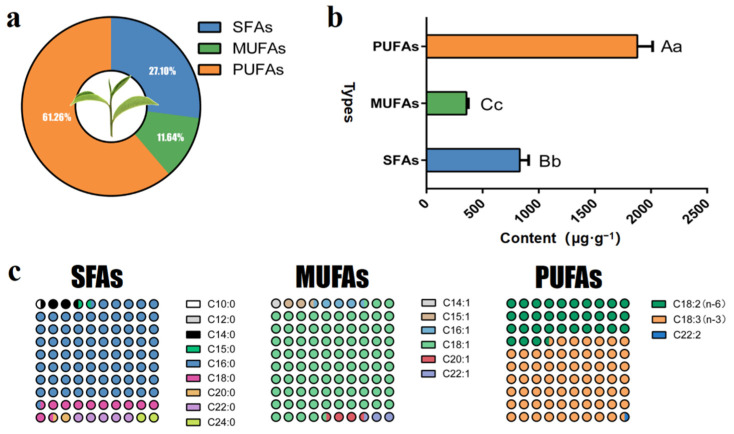
**Content and composition of fatty acids in fresh leaves of oolong tea cultivar** (**a**) Analysis of the content and difference of three kinds of fatty acids in fresh tea leaves. (**b**) Proportion of three types of fatty acids in fresh tea leaves. (**c**) Proportion of different carbon chain fatty acids in fresh tea leaves. Note: different uppercase letters (A–C) and lowercase alphabets (a–c) represent significant differences at *p* < 0.01 and *p* < 0.05, respectively.

**Figure 3 molecules-27-04298-f003:**
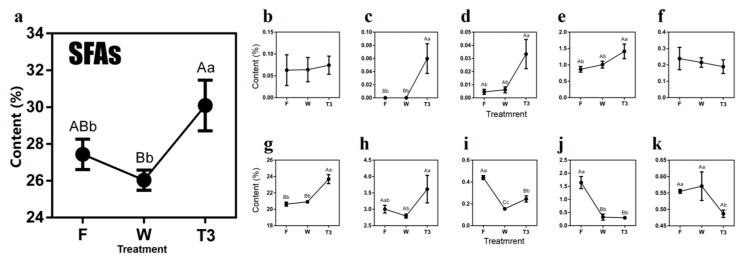
Dynamics of saturated fatty acids (SFAs) content during the postharvest process of oolong tea. (**a**) Overall trend of SFAs. (**b**) Methyl decanoate (C10:0). (**c**) Methyl undecanoate (C11:0). (**d**) Methyl laurate (C12:0). (**e**) Methyl myristate (C14:0). (**f**) C15:0(Methyl pentadecanoate). (**g**) C16:0 (Methyl palmitate). (**h**) Methyl stearate (C18:0). (**i**) Methyl arachidate (C20:0). (**j**) Methyl behenate (C22:0). (**k**) C24:0 (Methyl lignocerate). Note: different uppercase letters (A–C) and lowercase alphabets (a–c) represent significant differences at *p* < 0.01 and *p* < 0.05, respectively. F: fresh leaves; W: solar-withered leaves; T3: turn-over leaves.

**Figure 4 molecules-27-04298-f004:**
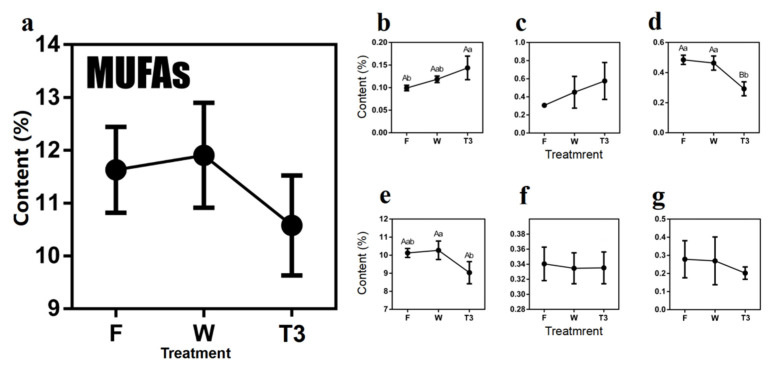
Dynamics of monounsaturated fatty acids (MUFAs) content during the postharvest process of oolong tea (**a**) overall trend of MUFAs. (**b**) Methyl myristoleate (C14:1). (**c**) Methyl *cis*-10-pentadecenoate (C15:1). (**d**) Methyl palmitoleate (C16:1). (**e**) Trans-9-Elaidic acid methyl ester (C18:1t). (**f**) Methyl *cis*-11-eicosenoate (C20:1(n-9)). (**g**) Methyl erucate (C22:1). Note: different uppercase letters (A, B) and lowercase alphabets (a, b) represent significant differences at *p* < 0.01 and *p* < 0.05, respectively. F: fresh leaves; W: solar-withered leaves; T3: turn-over leaves.

**Figure 5 molecules-27-04298-f005:**
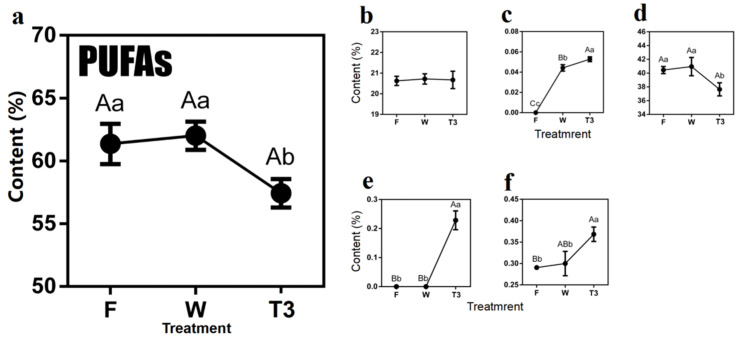
Dynamics of polyunsaturated fatty acids (PUFAs) content during the postharvest process of oolong tea. (**a**) Overall trend of PUFAs. (**b**) Methyl linolelaidate (C18:2(n-6c)). (**c**) Methyl γ-linolenate (C18:3(n-6)). (**d**) Methyl linolenate (α-C18:3). (**e**) *cis*-5,8,11,14,17-Eicosapentaenoic acid methyl ester (C20:5(n-3)). (**f**) *cis*-13,16-Docosadienoic acid methyl ester (C22:2). Note: different uppercase letters (A–C) and lowercase alphabets (a–c) represent significant differences at *p* < 0.01 and *p* < 0.05, respectively. F: fresh leaves; W: solar-withered leaves; T3: turn-over leaves.

**Figure 6 molecules-27-04298-f006:**
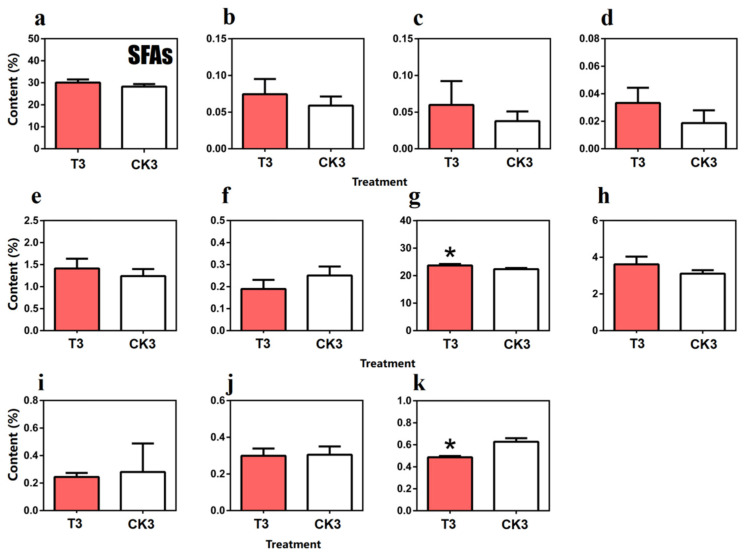
Comparison of saturated fatty acid (SFAs) content in turn-over leaves and spreading leaves during the postharvest process of oolong tea. (**a**) Overall trend of SFAs. (**b**) Methyl decanoate (C10:0). (**c**) Methyl undecanoate (C11:0). (**d**) Methyl laurate (C12:0). (**e**) Methyl myristate (C14:0). (**f**) C15:0(Methyl pentadecanoate). (**g**) C16:0 (Methyl palmitate). (**h**) Methyl stearate(C18:0). (**i**) Methyl arachidate(C20:0). (**j**) Methyl behenate(C22:0). (**k**) C24:0(Methyl lignocerate). Note: * represents significant differences at *p* < 0.05. T3: turn-over leaves; CK3: indoor-withered leaves.

**Figure 7 molecules-27-04298-f007:**
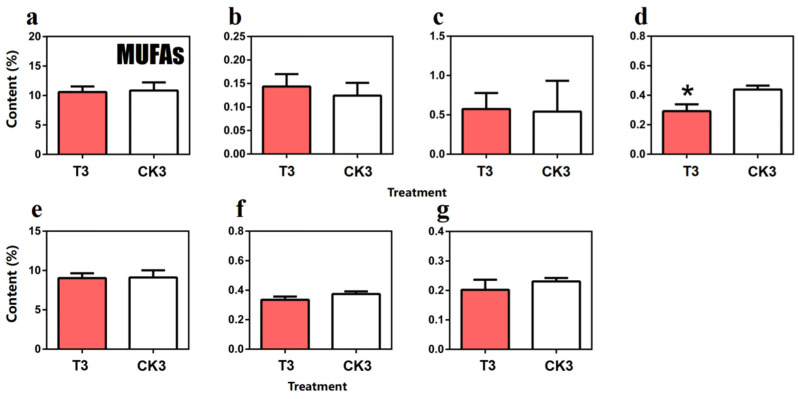
Comparison of monounsaturated fatty acids (MUFAs) content in turn-over leaves and spreading leaves during the postharvest process of oolong tea. (**a**) Overall trend of MUFAs. (**b**) Methyl myristoleate (C14:1). (**c**) Methyl *cis*-10-pentadecenoate (C15:1). (**d**) Methyl palmitoleate (C16:1). (**e**) Trans-9-Elaidic acid methyl ester (C18:1t). (**f**) Methyl *cis*-11-eicosenoate(C20:1(n-9)). (**g**) Methyl erucate(C22:1). Note: * represents significant differences at *p* < 0.05. T3: turn-over leaves; CK3: indoor-withered leaves.

**Figure 8 molecules-27-04298-f008:**
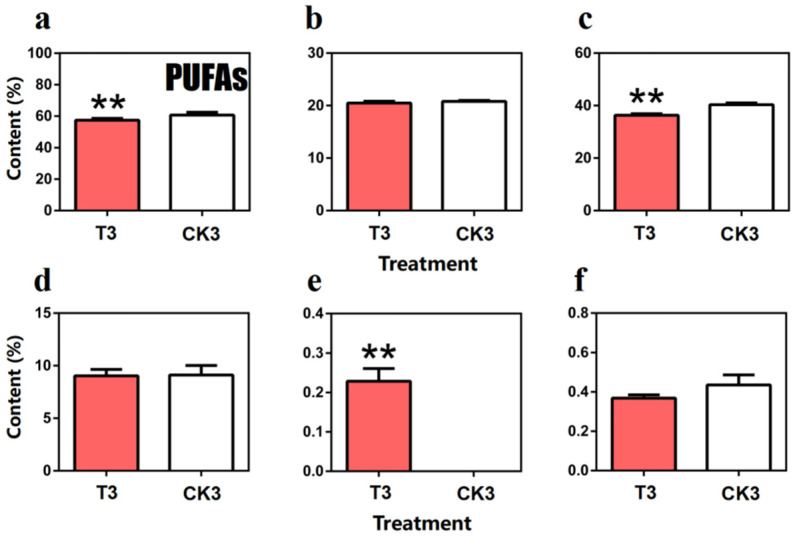
Comparison of polyunsaturated fatty acids (PUFAs) content in turn-over leaves and spreading leaves during the postharvest process of oolong tea. (**a**) Overall trend of PUFAs. (**b**) Methyl linolelaidate (C18:2(n-6c)). (**c**) Methyl γ-linolenate (C18:3(n-6)). (**d**) Methyl linolenate (α-C18:3). (**e**) *cis*-5,8,11,14,17-Eicosapentaenoic acid methyl ester (C20:5(n-3)). (**f**) *cis*-13,16-Docosadienoic acid methyl ester (C22:2). Note: ** represents extremely significant differences at *p* < 0.01. T3: turn-over leaves; CK3: indoor-withered leaves.

**Figure 9 molecules-27-04298-f009:**
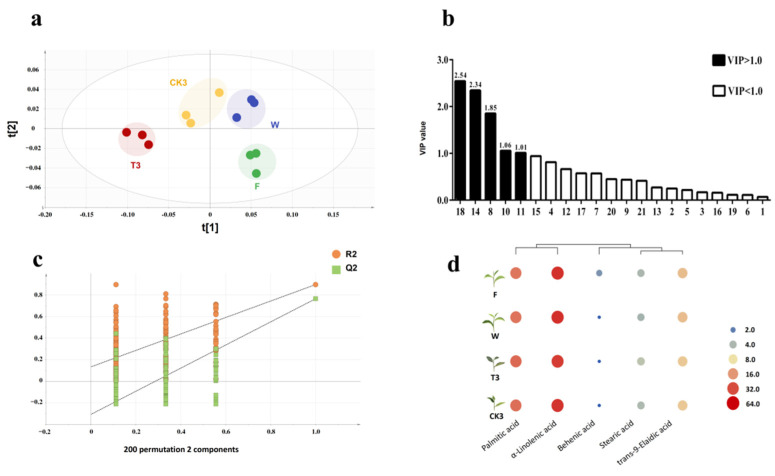
Partial least squares discriminant analysis (PLS-DA) model for key fatty acid components during the postharvest process of oolong tea production. (**a**) Partial least squares discrimination analysis (PLS-DA) score scatter plot. (**b**) Variable importance in projection (VIP) scores plot for PLS-DA plot. 1:Methyl decanoate(C10:0); 2: Methyl undecanoate (C11:0); 3: Methyl laurate (C12:0); 4: Methyl myristate (C14:0); 5: Methyl myristoleate (C14:1); 6: Methyl pentadecanoate (C15:0); 7: Methyl *cis*-10-pentadecenoate (C15:1); 8: Methyl palmitate(C16:0); 9: Methyl palmitoleate (C16:1); 10: Methyl stearate (C18:0); 11: Trans-9-Elaidic acid methyl ester (C18:1t); 12: Methyl linolelaidate (C18:2(n-6c)); 13: Methyl γ-linolenate (γ-C18:3); 14: Methyl linolenate (α-C18:3); 15: Methyl arachidate (C20:0); 16: Methyl *cis*-11-eicosenoate (C20:1(n-9)); 17: *cis*-5,8,11,14,17-Eicosapentaenoic acid methyl ester (C20:5(n-3)); 18: Methyl behenate (C22:0); 19: Methyl erucate (C22:1); 20: *cis*-13,16-Docosadienoic acid methyl ester (C22:2); 21: Methyl lignocerate (C24:0). (**c**) Permutation test analysis. (**d**) Heat map of key fatty acid components during the oolong tea production. Note: F: fresh leaves; W: solar-withered leaves; T3: turn-over leaves; CK3: indoor-withered leaves.

**Figure 10 molecules-27-04298-f010:**
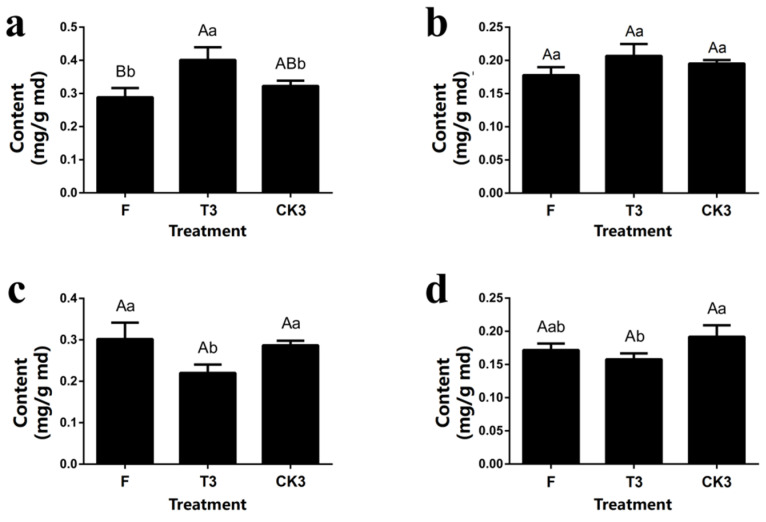
The content of four kinds of target fatty acids in samples during the postharvest process of oolong tea based on the UPLC-MS/MS method. (**a**) Palmitate acid. (**b**) Trans-9-Elaidic acid. (**c**) α-Linolenic acid. (**d**) Linoleic acid. Note: Different uppercase letters (A, B) and lowercase alphabets (a, b) represent significant differences at *p* < 0.01 and *p* < 0.05, respectively. F: fresh tea leaves; T3: turn-over leaves; CK3: indoor-withered leaves.

**Figure 11 molecules-27-04298-f011:**
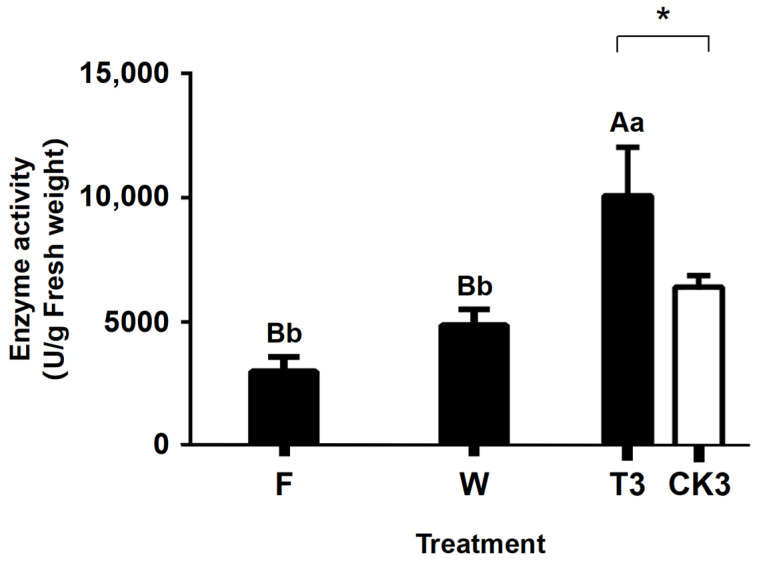
Changes of LOX (Lipoxygenase) activity during the postharvest process of oolong tea. Note: Different uppercase letters (A, B) and lowercase alphabets (a, b) represent significant differences at *p* < 0.01 and *p* < 0.05, respectively. * represents significant differences at *p* < 0.05. F: fresh tea leaves; W: solar-withered leaves; T3: turn-over leaves; CK3: indoor-withered leaves.

**Figure 12 molecules-27-04298-f012:**
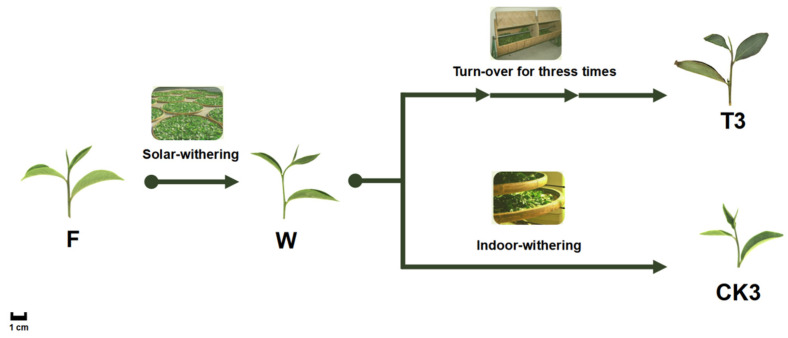
Apparent imaging and sampling diagrams during the postharvest process of oolong tea. The postharvest processing steps of fresh tea leaves. Solar withering made the fresh leaves (F) plucked from a tea plantation become withered leaves (W). Upstream was the experimental group consisting of turn-over and withering treatment of three times (T3), while the downstream was a control group with indoor-withering treatment (CK3); every sample was taken at the same time point of those in the experimental group.

**Table 1 molecules-27-04298-t001:** The retention time of fatty acids in oolong tea leaves.

Fatty Acid	Types	Retention Time (min)	Fatty Acid	Types	Retention Time (min)
Methyl decanoate	C10:0	2.029	Methyl linolelaidate	C18:2(n-6)	7.657
Methyl undecanoate	C11:0	2.381	Methyl γ-linolenate	C18:3(n-6)	8.015
Methyl laurate	C12:0	2.790	Methyl linolenate	C18:3(n-3)	8.434
Methyl myristate	C14:0	3.857	Methyl arachidate	C20:0	9.439
Methyl myristoleate	C14:1	4.128	Methyl *cis*-11-eicosenoate	C20:1(n-9)	9.671
Methyl pentadecanoate	C15:0	4.573	*cis*-5,8,11,14,17-Eicosapentaenoic acid methyl ester	C20:5(n-3)	12.785
Methyl *cis*-10-pentadecenoate	C15:1	4.744	Methyl behenate	C22:0	13.094
Methyl palmitate	C16:0	5.311	Methyl erucate	C22:1	13.252
Methyl palmitoleate	C16:1	5.516	*cis*-13,16-Docosadienoic acid methyl ester	C22:2	13.991
Methyl stearate	C18:0	6.933	Methyl lignocerate	C24:0	16.324
trans-9-Elaidic acid methyl ester	C18:1t	7.149			

**Table 2 molecules-27-04298-t002:** The correlation analysis between the contents of four types of fatty acids determined by GC-FID and UPLC-MS/MS.

	Palmitate (C16:0)		Trans-9-Elaidic Acid (C18:1t)		α-Linolenic Acid (C18:3(n-3))		Linoleic Acid (C18:2(n-6))	
	UPLC-MS/MS							
Four types of fatty acid content by GC-FID method	R	*p*	R	*p*	R	*p*	R	*p*
	0.863	**	−0.073	NS	0.800	**	0.699	*

Note: R: Pearson correlation coefficient. * and ** represent significant differences at *p* < 0.05 and *p* < 0.01, respectively. NS: non-significant.

## Data Availability

The original contributions presented in the study are included in the article/Appendix A, further inquiries can be directed to the corresponding author.

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
