# Peer review of "The Dynamic Change in Fatty Acids during the Postharvest Process of Oolong Tea Production"

_molecules, 2022, doi:10.3390/molecules27134298_

Round 1
Reviewer 1 Report
This manuscript analyzed the variation of PUFA, MUFA, and SCFA content in fresh and post-harvested tea leaves. The analytical procedures are well described, the methods are correct and the authors recorded a vast amount of data. The research work summarizes the gradual reduction of PUFA's content and its structural modifications of them. I have just a question: Which is the utility of this work? so, I mean, if the scientific pieces of evidence can find a resolution at a loss in PUFAs, this is a problem for the tea industry. We know that the industrial processes, could determine a reduction of the active substances, mainly in the tea leaves, in my opinion, research should have the aim to ameliorate a process. On the contrary, in this work, well done absolutely, there is an investigation but not a future perspective, on how these results can improve the oolong tea production, in terms of content quality.
Author Response
Thank you. This is a good question. Our work aims to explore how fatty acids(FAs) change during processing of oolong tea production, figuring out the sources of volatile fatty acid derivatives (VFADs), which not only helps us make a better understanding of formation mechanism of aroma of oolong tea production, but also can provide theoretical reference for the standardized production of oolong tea. The “ameliorate a process”you raised is very meaningful. We have already added the prospect that the appropriate increase of the turn-over process helps to the formation of tea aroma based on your point of view in the conclusion section. However, for different tea varieties and categories, increasing the intensity of turu-over, the length of time and the method of turn-over still need further research.
Aroma is a major factor in evaluating the quality of oolong tea, accounting for 30% of the quality weight. VFADs are one of the four aroma types of tea according the metabolic origins[1]. The post-harvest processing of fresh tea leaves can realize the formation process of tea flavor and quality, thereby enhancing the value of production[2]. As an important leaf-usage cash crop in the world, tea plant leaves are rich in fatty acid components and active oxidase[3]. The stimulation of exogenous turn-over mechanical force is a key factor to promote the oxidative degradation of FAs during the post-harvest of oolong tea[4]. And then derived a large number of volatile fatty acid aroma substances. Therefore, promoting the degradation of PUFAs is the key to forming the aroma of tea.
Ref:
[1] Yang, Z., Baldermann, S., Watanabe, N. Recent studies of the volatile compounds in tea. Food Res. Int. 2013, 53(2), 585-599.
[2] Feng, Z., Li, Y., Li, M., Wang, Y., Zhang, L., Wan, X., Yang, X. Tea aroma formation from six model manufacturing processes. Food Chem. 2019, 285,347-354.
[3] Zhou, Z., Wu, Q., Yao, Z., Deng, H., Liu B., Yue, C., Deng, T., Lai, Z., Sun Y. Dynamics of ADH and related genes responsible for the transformation of C6-aldehydes to C6-alcohols during the postharvest process of oolong tea. Food Sci & Nutr. 2020, 8, 104-113
[4] Zeng, L, Zhou, X., Su, X.,Yang, Z. Chinese oolong tea: An aromatic beverage produced under multiple stresses. Trends Food Sci. Tech. 2020, 106, 242-253.
Reviewer 2 Report
1. Some abbreviations such as ESI, MRM, FAME, should be adequately described.
2. Xylic acid (C24:0) should be Lignoceric acid.
Author Response
1. Some abbreviations such as ESI, MRM, FAME, should be adequately described.Response: Thank you for your reminder. We have made an adequate description.
L254: Electrospray ionization (ESI)
L254-255:Combined with multiple reaction monitoring(MRM)
L73-74:Fatty acid methyl esters(FAME)
2. Xylic acid (C24:0) should be Lignoceric acid.
Response: Thank you for your suggestion. We have replaced Xylic acid with Lignoceric acid.(L154-155; L321)
Reviewer 3 Report
Reviewer's comment on Manuscript Number: molecules-1787947
I am writing in reference to the manuscript entitled "The dynamic change of fatty acids during the post-harvest process of oolong tea production”.
The paper is focused on the transformation of fatty acids during the process of oolong tea production.The subject of the manuscript falls within the scope of Molecules. The results are certainly very interesting and valuable. This study could stimulate more research to further the exploration of the problem.
However, I have specific comments regarding the manuscript: 1. The paper should be checked by a native English speaker. There are several errors that need attention. 2. In general paper lacks validation of each parameter analysis. It should be corrected. 3. Abbreviations should be explained when first used. It is difficult to read introduction as there are many abbreviations without explanation. 4. Stastitical part of the manuscript lacks discussion of results. Therefore, I propose to accept this paper for publication in Molecules after major amendments.Author Response
1. The paper should be checked by a native English speaker. There are several errors that need attention.
Response: Thank you for your reminder. We are really sorry about our carelessness. We have invited a native English speaker to help us check the error. According to her revision comments, we have completed the revision and improvement of the manuscript. If necessary, we don’t mind purchase a extensive language service to present a better manuscript.
2. In general paper lacks validation of each parameter analysis. It should be corrected.
Response: Thank you for your suggestion. In fact, Aiming at the main results of GD-FID, we carried out the experimental verification of UPLC-MS/MS method. By comparing the two methods, we strive to ensure that our detection results are convincing in multiple dimensions(L241-260). In addition, in the construction of the PLS-DA model, we also have method validation for this model to ensure the accuracy of the analysis results.(Figure. 10-C)
Ref:
[1] Zhou, Z.W.; You, F.N.; Liu, B.B.; Deng, T.T.; Lai, Z.X.; Sun, Y. Effect of mechanical force during turning-over on the formation of aliphatic aroma in oolong Tea, Food Sci. 2019, 40(13), 52-59.
[2] DovilÄ—, J., Joana, Š., Joana, Š., Galina, G., Gintare, Z. Determination of fatty acid composition in Bacillus cereus isolated from dried milk products. Milchwissenschaft 2012, 67(4), 377-380.
3. Abbreviations should be explained when first used. It is difficult to read introduction as there are many abbreviations without explanation.
Response: I apologize for our carelessness. Thank you for your reminder We have made an adequate description.
L74: Fatty acid methyl esters(FAME)
L84: saturated fatty acid (SFAs),
L84: momounsatturated fatty acid (MUFAs)
L85: polyunsaturated fatty acid (PUFAs)
L254: Electrospray ionization (ESI)
L254-255:Combined with multiple reaction monitoring(MRM)
L256: declustering potential (DP)
L257: collision energy (CE)
4. Stastitical part of the manuscript lacks discussion of results.
Response:Thanks for your suggestion. We have already added discussion OF Stastitical part results in L399-346。
“Based on the differences in the content of FAs components in the samples, we established a PLS-DA model to preliminarily distinguish samples during the post-harvest processing of oolong tea production[30]. According to the discrete degree of F_v_T3 and F_v_CK3 in the load diagram, we could infer that the participation of the mechanical force is an important factor in promoting the change of the principal components of FAs[31].The FAs screened under the condition that vip >1.0 reflected that the change of fatty acid carbon chain is more obvious than the change of saturated bond[32-33]”.
Ref.
[30] Wang, M., Shao, C., Zhu, Y., Zhang, Y., Lin, Z., Lyu, H. Aroma constituents of Longjing tea produced in different areas. J. Tea Sci. 2018, 38(05), 508-517.
[31] Gui, J., Fu, X., Zhou, Y., Katsuno, T., Mei, Xin., Deng, R., Xu, X., Zhang, L., Dong, F., Watanabe, N., Yang, Z. Does Enzymatic Hydrolysis of Glycosidically Bound Volatile Compounds Really Contribute to the Formation of Volatile Compounds During the Oolong Tea Manufacturing Process?. J. agri. food chem. 2015, 63(31), 6905-6914.
[32] Ni, Z., Wu, Q., Zhou, Z., Yang ,Y., Hu, Q., Deng, H., Zheng, Y., Bi, W., Liu, Z., Sun, Y. Effects of turning over intensity on fatty acid metabolites in postharvest leaves of Tieguanyin oolong tea(Camellia sinensis) PeerJ 2022, 10, e13453.
[33] Guo, L., Chen, M., Guo Y., Lin Z. Variations in fatty acids affected their derivative volatiles during Tieguanyin tea processing. Foods 2022, 11(11), 1563.
Round 2
Reviewer 3 Report
The paper was improved. It can be accepted.